# Comparing Several Gamma Means: An Improved Log-Likelihood Ratio Test

**DOI:** 10.3390/e25010111

**Published:** 2023-01-05

**Authors:** Augustine Wong

**Affiliations:** Department of Mathematics and Statistics, York University, Toronto, ON M3J 1P3, Canada; august@yorku.ca

**Keywords:** Bartlett correction, homogeneity of means, *p*-value

## Abstract

The two-parameter gamma distribution is one of the most commonly used distributions in analyzing environmental, meteorological, medical, and survival data. It has a two-dimensional minimal sufficient statistic, and the two parameters can be taken to be the mean and shape parameters. This makes it closely comparable to the normal model, but it differs substantially in that the exact distribution for the minimal sufficient statistic is not available. A Bartlett-type correction of the log-likelihood ratio statistic is proposed for the one-sample gamma mean problem and extended to testing for homogeneity of k≥2 independent gamma means. The exact correction factor, in general, does not exist in closed form. In this paper, a simulation algorithm is proposed to obtain the correction factor numerically. Real-life examples and simulation studies are used to illustrate the application and the accuracy of the proposed method.

## 1. Introduction

Consider a sample of $\left(x_{1},\ldots ,x_{n}\right)$ from the two-parameter gamma model with mean μ and shape λ. The joint density is
(1)$$\begin{array}{c} f\left(x_{1},\ldots ,x_{n};\mu ,\lambda \right)=\Gamma^{-n}\left(\lambda \right){\left(\frac{\lambda }{\mu }\right)}^{n\lambda }\exp\left\{\lambda t-\frac{\lambda }{\mu }s\right\}\prod_{i=1}^{n}\frac{1}{x_{i}} \end{array}$$
where $\left(s,t\right)=\left(\sum_{i=1}^{n}x_{i},\sum_{i=1}^{n}\log x_{i}\right)$ is a minimal sufficient statistic. The two-parameter gamma distribution is often used to model the non-negative data with right-skewed distribution. Moreover, depending on the values of the parameters, it can have a decreasing failure rate, a constant failure rate, or an increasing failure rate. This makes it a valuable model for analyzing data arising from engineering, environmental, meteorological, and medical studies.

Similar to the normal distribution, the two-parameter gamma distribution has a two-dimensional minimal sufficient statistic (s,t). Another version of the minimal sufficient statistic is (r,s), where *r* is the log offset of the arithmetic mean from the geometric mean
$$r=\log\frac{\sum_{i=1}^{n}x_{i}}{n}-\log{\left(\prod_{i=1}^{n}x_{i}\right)}^{1/n}=\log\frac{s}{n}-\frac{t}{n}.$$ Notice that the density of $\log x_{i}$ has location model form for a fixed λ. It follows from [1] that the conditional density for *s* given *r* and the marginal density for *r* take the form
$$\begin{array}{ccc} f(r;\lambda ) & = & \Gamma \left(n\lambda \right)\Gamma^{-n}\left(\lambda \right)n^{-n\lambda +1/2}\exp\left\{-n\lambda r\right\}h_{n}\left(r\right)\\ f(s|r;\mu ,\lambda ) & = & \Gamma^{-1}\left(n\lambda \right){\left(\frac{\lambda }{\mu }\right)}^{n\lambda }\exp\left\{n\lambda \log s-\frac{\lambda s}{\mu }\right\}\frac{1}{s}, \end{array}$$
where $h_{n}\left(r\right)$ appears in the transformed measure
$$\prod_{i=1}^{n}\frac{1}{x_{i}}dx_{i}=\frac{1}{s}ds\times \sqrt{n}h_{n}\left(r\right)dr$$
which requires (n−2)-dimensional integration. Hence, the joint density for (s,r) is
f(s,r;μ,λ)=f(s|r;μ,λ)f(r). By change of variable, the joint density for (s, t) takes the form
(2)$$\begin{array}{c} f\left(s,t;\mu ,\lambda \right)=\Gamma^{-n}\left(\lambda \right){\left(\frac{\lambda }{\mu }\right)}^{n\lambda }\exp\left\{\lambda t-\frac{\lambda }{\mu }s\right\}\frac{1}{s}\frac{1}{\sqrt{n}}h_{n}\left(\log\frac{s}{n}-\frac{t}{n}\right). \end{array}$$ The same result can be obtained by using the properties of the exponential transformation model by [2] or the conditional argument in [3]. Note that $h_{n}\left(\cdot \right)$ requires (n−2)-dimensional integration, and it is available exactly only for small value of *n* (see [3,4]). Unlike the normal model, where inference for the normal mean can be obtained explicitly, inference for the gamma mean is a complicated and challenging problem. Many asymptotics inferential methods for the gamma mean exist in statistical literature. Moreover, most of the existing asymptotics methods are likelihood-based methods. Some are very simple but not very accurate, and others are very accurate but mathematically complicated and also computationally intensive. Furthermore, only limited methods can be applied to the problem of comparing the means of k>2 independent gamma distributions.

Ref. [5] considered the log-likelihood function obtained from (1), which takes the form
(3)$$\begin{array}{c} \ell \left(\mu ,\lambda \right)=-n\log\Gamma \left(\lambda \right)+n\lambda \log\lambda -n\lambda \log\mu +\lambda t-\frac{\lambda }{\mu }s. \end{array}$$ The maximum likelihood estimate (MLE) is $\left(\hat{\mu },\hat{\lambda }\right)$, where $\hat{\mu }=\frac{s}{n}$, and $\hat{\lambda }$ satisfies
$$-\psi \left(\hat{\lambda }\right)+\log\hat{\lambda }-\log\frac{s}{n}+\frac{t}{n}=0$$
with ψ(·) being the digamma function. In addition, the observed information matrix evaluated at MLE is
$$\hat{j}=\left(\begin{array}{cc} -\frac{n\hat{\lambda }}{{\hat{\mu }}^{2}}+\frac{2\hat{\lambda }}{{\hat{\mu }}^{3}}s & \frac{n}{\hat{\mu }}-\frac{s}{{\hat{\mu }}^{2}}\\ \frac{n}{\hat{\mu }}-\frac{s}{{\hat{\mu }}^{2}} & -\frac{n}{\hat{\lambda }}+n\psi^{\prime }\left(\hat{\lambda }\right) \end{array}\right).$$
It is well-known that the variance-covariance matrix of the maximum likelihood estimators can be approximated by ${\hat{j}}^{-1}$. Hence, the standardized maximumm likelihood estimator is
$$\begin{array}{c} Q\left(\mu \right)=\frac{\hat{\mu }-\mu }{\sqrt{\hat{var}\left(\hat{\mu }\right)}}, \end{array}$$
where $\hat{var}\left(\hat{\mu }\right)$ can be approximated by the (1,1) entry of ${\hat{j}}^{-1}$. With the regularity conditions stated in [6] and also in [7], for large *n*,
(4)$$\begin{array}{c} Q\left(\mu \right)\overset{d}{⟶}N\left(0,1\right) \end{array}$$
with first order accuracy, $O(n^{-1/2})$. Thus, inference for μ can be obtained based on the limiting distribution. This method is generally known as the Wald method or the asymptotic MLE method.

Another commonly used method to obtain inference for μ is the log-likelihood ratio method. Let ${\tilde{\lambda }}_{\mu }$ be the constrained MLE, which maximizes ℓ(μ,λ) for a fixed μ. In this case, ${\tilde{\lambda }}_{\mu }$ must satisfy
$$-\psi \left({\tilde{\lambda }}_{\mu }\right)+\log{\tilde{\lambda }}_{\mu }-\log\mu +1-\frac{s}{n\mu }+\frac{t}{n}=0.$$
Then the log-likelihood ratio statistic is
$$\begin{array}{c} W\left(\mu \right)=2[\ell \left(\hat{\mu },\hat{\lambda }\right)-\ell \left(\mu ,{\tilde{\lambda }}_{\mu }\right)]. \end{array}$$
Again, with the regularity conditions stated in [6] and also in [7], for large *n*,
(5)$$\begin{array}{c} W\left(\mu \right)\overset{d}{⟶}\chi_{1}^{2}. \end{array}$$
with first order accuracy, $O(n^{-1/2})$. Hence, inference for μ can be approximated based on the limiting $\chi_{1}^{2}$ distribution. This method is also known as the Wilks method.

To improve the accuracy of the log-likelihood ratio method, ref. [8] applied the Bartlett correction to the log-likelihood ratio statistic (see [9]). The resulting Bartlett corrected log-likelihood ratio statistic takes the form
(6)$$\begin{array}{c} W^{*}\left(\mu \right)=\frac{W(\mu )}{1+B(\mu )/n} \end{array}$$
where
$$\begin{array}{ccc} B(\mu ) & = & \frac{1}{6{\tilde{\lambda }}_{\mu }}+\frac{1}{2}\frac{D_{3}\left({\tilde{\lambda }}_{\mu }\right)}{{\left[D_{2}\left({\tilde{\lambda }}_{\mu }\right)\right]}^{2}}+\frac{1}{4D_{2}\left({\tilde{\lambda }}_{\mu }\right)}\\ D_{2}\left(\lambda \right) & = & \lambda^{2}\left[\lambda^{-1}-\psi^{\prime }\left(\lambda \right)\right]\\ D_{3}\left(\lambda \right) & = & \lambda^{3}\left[-\lambda^{-2}-\psi^{\prime \prime }\left(\lambda \right)\right], \end{array}$$
and B(·) is known as the Bartlett correction factor. Ref. [9] showed that the Bartlett corrected log-likelihood ratio statistic converges to $\chi_{1}^{2}$ distribution with fourth order accuracy. Hence, inference for μ can be approximated based on the limiting $\chi_{1}^{2}$ distribution.

Refs. [3,4] showed that the exact form of $h_{n}\left(\cdot \right)$ in (2) is only available when *n* is small. Jensen used the fact that the model is an exponential-transformation model and applied the saddlepoint method to approximate $h_{n}\left(\cdot \right)$ and derived an inference procedure for μ with third order accuracy. However, due to the complexity of the method, ref. [3] only provided tables for 1, 2.5, 97.5, and 99 percentiles of μ for sample sizes 10, 20, 40, and *∞*, which were obtained by extensive iterative calculations. On the other hand, ref. [4] proposed another third order method to obtain inference for μ. This method is asymptotically equivalent to Jensen’s method with the exception that it involves direct implementation of the method derived in [10].

Note that Gross and Clark’s method is very simple but not very accurate. The log-likelihood ratio method is slightly more complicated because of the calculation of the constrained MLE, and it is still not very accurate. The Bartlett corrected log-likelihood ratio method by [8] gives very accurate results. It is also relatively straightforward because [8] derived all the necessary equations. The method presented in [4] is also very accurate, but it is computational intensive. Gross and Clark’s method, Jensen’s method and Fraser, Reid and Wong’s method are not applicable to the problem of testing homogeneity of k>2 independent gamma means. The log-likelihood ratio method can be extended to this problem but it has only first order accuracy. Ref. [8] also derived the explicit Bartlett correction factor for testing equality of two independent gamma means, but, due to the complexity of the method, did not derive the explicit Bartlett correction factor for testing homogeneity of k>2 independent gamma means.

In this paper, a Bartlett-type correction of the log-likelihood ratio statistic is proposed in Section 2. The proposed Bartlett-type correction factor is numerically obtained by simulations. The proposed method is then applied testing homogeneity of k≥2 independent gamma means in Section 3 and Section 4, respectively. Some concluding remarks are given in Section 5. Real-life examples and simulation studies results are presented to compare the accuracy of the proposed method with the existing methods.

## 2. Main Results

Let ℓ(θ) be the log-likelihood function with a *p*-dimensional parameter θ. With the regularity conditions stated in [6], the log-likelihood ratio statistic,
$$W\left(\theta \right)=2[\ell \left(\hat{\theta }\right)-\ell \left(\theta \right)],$$
is asymptotically distributed as $\chi_{p}^{2}$ with first order accuracy, where $\hat{\theta }$ is the overall MLE, which maximizes ℓ(θ). Ref. [9] showed that the mean of W(θ) can be expressed as
$$E\left[W\left(\theta \right)\right]=p\left[1+\frac{B(\theta )}{n}+O\left(n^{-2}\right)\right],$$
where *n* is the size of the observed sample and B(·) is the Bartlett correction factor. Hence, the Bartlett corrected log-likelihood ratio statistic is
$$W^{*}\left(\theta \right)=\frac{W(\theta )}{1+B(\theta )/n}\overset{d}{⟶}\chi_{p}^{2}$$
and has mean *p* with fourth order accuracy.

The above method can be generalized to the case when ψ=ψ(θ) is the parameter of interest and dimension of ψ is m<p. With the regularity conditions stated in [6], the log-likelihood ratio statistic
$$W\left(\psi \right)=2[\ell \left(\hat{\theta }\right)-\ell \left({\tilde{\theta }}_{\psi }\right)]$$
is asymptotically distributed as $\chi_{m}^{2}$ with first order accuracy. Note that $\hat{\theta }$ is the overall MLE, which maximizes ℓ(θ), and ${\tilde{\theta }}_{\psi }$ is the constrained MLE, which maximizes ℓ(θ) for a given value of ψ. The Bartlett corrected log-likelihood ratio statistic is then
$$W^{*}\left(\psi \right)=\frac{W(\psi )}{1+B(\psi )/n}\overset{d}{⟶}\chi_{m}^{2}$$
and $W^{*}\left(\psi \right)$ has mean *m* with fourth order accuracy.

Theoretically, the Bartlett correction method gives extremely accurate results, even for small sample sizes. However, obtaining the explicit closed form expression of the Bartlett correction factor is a very difficult problem. There exists only limited problems in statistical literature that the explicit closed form, or even the asymptotic form of the Bartlett correction is available. For example, Ref. [8] obtained the Bartlett correction factor for the one-sample gamma mean problem as well as for the equality of the two independent gamma means problem only. However, they did not discuss the case for testing homogeneity of k>2 independent gamma means. The aim of this paper is to propose a systematic way of approximating a Bartlett-type correction factor.

Since the log-likelihood ratio statistic W(ψ) has the limiting distribution $\chi_{m}^{2}$, similar to the Bartlett correction method, we want to find a scale transformation of W(ψ) such that the transformed statistic has the exact mean *m*. An obvious transformation is
$$W^{†}\left(\psi \right)=\frac{W(\psi )}{E[W(\psi )]/m}.$$
However, calculating E[W(ψ)] is extremely complicated, if not impossible. If we have an observed sample of W(ψ), we can then apply the method of moments to estimate E[W(ψ)]. The primary task is to obtain such a sample. We propose to employ simulations to create such a sample. The main idea to to generate samples from the original model but with the parameters being the constrained MLE obtained from the original observed sample. We summarized the idea into the following algorithm.
Assume:$\left(x_{1},\cdots ,x_{n}\right)$ is a sample from a model with density
f(·;θ), and ψ is the parameter of interest.Have:The log-likelihood function ℓ(θ) is given in (3).
From the log-likelihood function, we can obtain $\hat{\theta },{\tilde{\theta }}_{\psi },$
and W(ψ).Step 1:Generate a sample of size $n,(x_{1}^{s},\cdots ,x_{n}^{s})$, from
the density $f(\cdot ;{\tilde{\theta }}_{\psi })$.Step 2:From the simulated sample, obtain the log-likelihood ratio statistic and
denote it as $W^{s}\left(\psi \right)$.Step 3:Repeat Steps 1 to 2 *N* times, where *N* is large.
As a result, we have $\left(W_{1}^{s}\left(\psi \right),\cdots ,W_{N}^{s}\left(\psi \right)\right)$.Step 4:Obtain
$${\bar{W}}^{s}\left(\psi \right)=\frac{\sum_{i=1}^{N}W_{i}^{s}\left(\psi \right)}{N}.$$Step 5:By the method of moments, ${\bar{W}}^{s}\left(\psi \right)$ is an consistent estimate of
E[W(ψ)]. Hence,
$$W^{†}\left(\psi \right)=\frac{W(\psi )}{{\bar{W}}^{s}\left(\psi \right)/m}$$
has mean *m*, and its limiting distribution is $\chi_{m}^{2}$.

## 3. One-Sample Gamma Mean Problem

Consider the one-sample gamma mean problem with the log-likelihood given in (3). Ref. [5] proposed to use Wald statistic given in (4) to obtain inference for μ, whereas [8] recommended to use the Bartlett corrected log-likelihood ratio statistic given in (6) to obtain inference for μ. In this paper, a Bartlett-typed corrected log-likelihood ratio statistic based on the algorithm given in Section 2 is proposed as an alternative approach to obtain inference for μ. To compare the results obtained by these methods, we consider the data set given in [5], which is the survival time of 20 mice exposed to 240 rad of gamma radiation:
152 152 115 109 137 88 94 77 160 165125 40 128 123 136 101 62 153 83 69

Table 1 recorded the 95% confidence intervals for μ and the *p*-values for testing $H_{0}:\mu =133$ vs. $H_{a}:\mu \neq 133$ obtained by the methods discussed in this paper. From Table 1, we observed that results obtained by Jensen and Kristensen’s method and by the proposed method are almost identical. However, they are very different from the results obtained by Gross and Clark’s method and the standard log-likelihood ratio statistic method.

Simulation studies are performed to compare the accuracy of the methods discussed in this paper. In particular, 5000 simulated samples are obtained for each combination of μ,λ, and *n*. Moreover, we use N=100 for each of the simulated sample to approximate the mean of the log-likelihood ratio statistic. The proportion of samples that was rejected at 5% significance level are recorded in Table 2. Theoretically, the true percentage of samples that will be rejected is 5% with a standard deviation of 0.31%. Extensive simulation studies were performed. All results are very similar and, therefore, only a subset of them are presented in Table 2. Results from other combinations of the parameters are available from the authors upon request.

We observed that results by Gross and Clark’s method are significantly different from the nominal 5% level, but the accuracy is improving slowly as the sample size *n* increases. Results by the log-likelihood ratio method are slightly better, and the accuracy improves much faster as the sample size increases. Results by both Jensen and Kristensen’s method and the proposed method are very accurate and always within 3 standard deviation of the nominal 5% level, even when the sample size is as small as 5.

## 4. Testing Homogeneity of k Independent Gamma Means

In this section, the proposed method is extended to testing homogeneity of k≥2 independent gamma means problem. Let $\left(x_{i1},\ldots x_{in_{i}}\right)$ be a sample from the two-parameter gamma model with mean $\mu_{i}$ and shape $\lambda_{i}$, where i=1,…,k and k≥2. Moreover, assume the *k* two-parameter gamma models are independent. Let $\ell_{i}\left(\mu_{i},\lambda_{i}\right)$ be the log-likelihood function from the *i*th model. Then the joint log-likelihood function is
(7)$$\begin{array}{ccc} \ell (\mu_{1},\ldots ,\mu_{k},\lambda_{1},\ldots ,\lambda_{k}) & = & \sum_{i=1}^{k}\ell_{i}\left(\mu_{i},\lambda_{i}\right)\\ & = & \sum_{i=1}^{k}\left\{n_{i}\lambda_{i}\log\lambda_{i}-n_{i}\lambda_{i}\log\mu_{i}-n_{i}\log\Gamma \left(\lambda_{i}\right)+\lambda_{i}t_{i}-\frac{\lambda_{i}}{\mu_{i}}s_{i}\right\} \end{array}$$
where
$$s_{i}=\sum_{j=1}^{n_{i}}x_{ij}\;\mathrm{and}\;t_{i}=\sum_{j=1}^{n_{i}}\log x_{ij}.$$ Since the *k* models are independent, the overall MLE is $\left({\hat{\mu }}_{1},\ldots ,{\hat{\mu }}_{k},{\hat{\lambda }}_{1},\ldots ,{\hat{\lambda }}_{k}\right)$ where ${\hat{\mu }}_{i}=\frac{s_{i}}{n_{i}}$ and ${\hat{\lambda }}_{i}$ must satisfy
$$-\psi \left({\hat{\lambda }}_{i}\right)+\log\left({\hat{\lambda }}_{i}\right)-\log\frac{s_{i}}{n_{i}}+\frac{t_{i}}{n_{i}}=0.$$ Hence, the log-likelihood function is evaluated at MLE $\ell ({\hat{\mu }}_{1},\ldots ,{\hat{\mu }}_{k},{\hat{\lambda }}_{1},\ldots ,{\hat{\lambda }}_{k})$.

To test homogeneity of *k* gamma means, the null and alternative hypotheses are
$$H_{0}:\mu_{1}=\cdots =\mu_{k}=\mu \;\mathrm{vs}.\;H_{a}:\mathrm{not}\;\mathrm{all}\;\mathrm{equal}.$$ The constrained log-likelihood function is
(8)$$\begin{array}{ccc} \ell (\mu ,\ldots ,\mu ,\lambda_{1},\ldots ,\lambda_{k}) & = & \sum_{i=1}^{k}\left\{n_{i}\lambda_{i}\log\lambda_{i}-n_{i}\lambda_{i}\log\mu -n_{i}\log\Gamma \left(\lambda_{i}\right)+\lambda_{i}t_{i}-\frac{\lambda_{i}}{\mu }s_{i}\right\}. \end{array}$$ The constrained MLE can must satisfy
$$\tilde{\mu }=\frac{\sum_{i=1}^{k}s_{i}{\tilde{\lambda }}_{i}}{\sum_{i=1}^{k}n_{i}{\tilde{\lambda }}_{i}}\;\mathrm{and}\;-\psi \left({\tilde{\lambda }}_{i}\right)+\log\left({\tilde{\lambda }}_{i}\right)+1-\log\tilde{\mu }-\log\frac{s_{i}}{n_{i}\tilde{\mu }}+\frac{t_{i}}{n_{i}}=0.$$ Thus, the log-likelihood function evaluated at the constrained MLE is $\ell (\tilde{\mu },\ldots ,\tilde{\mu },{\tilde{\lambda }}_{1},\ldots ,{\tilde{\lambda }}_{k})$. Finally, the log-likelihood ratio statistic is
$$W\left(\tilde{\mu }\right)=2\left[\ell \left({\hat{\mu }}_{1},\ldots ,{\hat{\mu }}_{k},{\hat{\lambda }}_{1},\ldots ,{\hat{\lambda }}_{k}\right)-\ell \left(\tilde{\mu },\ldots ,\tilde{\mu },{\tilde{\lambda }}_{1},\ldots ,{\tilde{\lambda }}_{k}\right)\right],$$
and it is asymptotically distributed as $\chi_{k-1}^{2}$ with first order accuracy. Note that for this problem, the Wald method is not applicable. Moreover, ref. [8] did not derive the exact Bartlett correction factor for k>2. However, the proposed method is still applicable.

To illustrate the application of the log-likelihood ratio method and the proposed method for this problem, we consider the intervals in service-hours between failures of the air-conditioning equipment in 10 Boeing 720 jet aircrafts reported in “Example T” from [11]. It is assumed that the reported times for each aircraft is distributed as a two-parameter gamma distribution. The question of interest is whether the ten aircrafts have the same mean intervals in service hours between failure. In other words, we are testing
$$H_{0}:\mu_{1}=\cdots =\mu_{10}=\mu \;\mathrm{vs}.\;H_{a}:\mathrm{not}\;\mathrm{all}\;\mathrm{equal}.$$ The log-likelihood ratio method gives a *p*-value of 0.0871, whereas the proposed method gives a *p*-value of 0.1295 (using *N* = 100,000). At 10% level of significance, the two methods give contradictory results with the log-likelihood ratio method rejecting $H_{0}$, and the proposed method failing to reject $H_{0}$.

As in the one sample case, to compare the accuracy of the two methods, extensive simulation studies were performed. For each combination of $k,n_{1},\cdots ,n_{k},\left(\mu_{1},\lambda_{1}\right),\cdots ,\left(\mu_{k},\lambda_{k}\right)$, 5000 simulated samples are obtained. Moreover, for each simulated sample, N = 100 is used to estimate the mean of the log-likelihood ratio statistic. The proportion of samples that reject the null hypothesis of homogeneity of the *k* means at 5% significance levels are recorded. Since all results are very similar, only results from the 8 cases listed in Table 3 are reported.

Results in Table 4 and Table 5 demonstrate that the log-likelihood ratio method gives unsatisfactory results, especially when the sample sizes are small. However, the accuracy of the results improve as the sample sizes increase. In comparison, the results from the proposed method are very accurate, and they are always within 3 standard deviation of the nominal 5% level, regardless of the sample sizes.

## 5. Conclusions

In this paper, a Bartlett-type corrected log-likelihood ratio method for comparing the means of several independent gamma distributions is proposed. The method can easily be applied in statistics software, such as *R*. Simulation results demonstrate the log-likelihood ratio method does not give satisfactory results, especially when the sample sizes are small. However, the proposed method is extremely accurate even when the sample sizes are small. One advantage of the proposed method is that it is not restricted to the gamma means problem, as it is applicable to any parametric models.

## Figures and Tables

**Table 1 entropy-25-00111-t001:** 95% confidence interval for μ and *p*-value for testing $H_{0}:\mu =133$ vs. $H_{a}:\mu \neq 133$.

Method	Confidence Interval	*p*-Value
Gross and Clark	(96.7, 130.2)	0.0223
Log-likelihood ratio	(97.6, 133.0)	0.0499
Jensen and Kristensen	(97.0, 133.9)	0.0586
Proposed	(97.0, 133.9)	0.0596

**Table 2 entropy-25-00111-t002:** Observed proportion of samples (in percentage) that was rejected $\mu =\mu_{0}$ at 5% significance levels of 5000 simulated samples.

$\mu_{0}$	$\lambda_{0}$	*n*	Gross and Clark	Log-Likelihood Ratio	Jensen and Kristensen	Proposed
1.5	0.5	5	21.82	9.70	5.70	5.20
		10	14.56	6.82	5.12	5.04
		15	11.84	5.94	4.68	4.98
		20	10.90	5.98	5.10	5.30
2.0	2.0	5	17.42	9.36	5.76	5.02
		10	10.54	7.42	5.24	5.64
		15	9.08	6.64	5.62	5.78
		20	7.38	5.54	4.62	5.04
4.0	1.0	5	18.76	9.26	5.48	5.02
		10	12.66	6.96	5.04	5.10
		15	10.80	6.20	5.16	5.32
		20	8.76	5.60	4.66	4.76

**Table 3 entropy-25-00111-t003:** Various combinations of $k,\left(\mu_{1},\lambda_{1}\right),\ldots ,\left(\mu_{k},\lambda_{k}\right)$.

Case	*k*	$\left(\mu_{1},\lambda_{1}\right),\cdots ,\left(\mu_{k},\lambda_{k}\right)$
(1)	3	(2,2),(2,2),(2,2)
(2)	3	(1,2),(1,4),(1,5)
(3)	3	$\left(3,\frac{1}{3}\right),\left(3,\frac{2}{3}\right),\left(3,\frac{4}{3}\right)$
(4)	3	$\left(1,1\right),\left(1,\frac{1}{2}\right),\left(1,\frac{1}{3}\right)$
(5)	5	(2,2),(2,2),(2,2),(2,2),(2,2)
(6)	5	(1,2),(1,3),(1,5),(1,7),(1,8)
(7)	5	$\left(1,\frac{1}{3}\right),\left(1,\frac{1}{2}\right),\left(1,\frac{1}{4}\right),\left(1,\frac{3}{4}\right),\left(1,\frac{2}{3}\right)$
(8)	5	$\left(2,1\right),\left(2,3\right),\left(2,\frac{1}{4}\right),\left(2,\frac{2}{3}\right),\left(2,2\right)$

**Table 4 entropy-25-00111-t004:** Observed proportion of samples (in percentage) that rejected the null hypothesis of homogeneity of means at 5% significance levels.

Case (1)				
$n_{1}$	$n_{2}$	$n_{3}$	Log-likelihood ratio	Proposed
5	5	5	0.1082	0.0490
5	8	15	0.0920	0.0544
8	12	16	0.0732	0.0496
10	15	20	0.0692	0.0542
30	30	30	0.0592	0.0552
Case (2)				
$n_{1}$	$n_{2}$	$n_{3}$	Log-likelihood ratio	Proposed
5	5	5	0.1090	0.0532
5	8	15	0.0998	0.0546
8	12	16	0.0738	0.0546
10	15	20	0.0706	0.0552
30	30	30	0.0618	0.0554
Case (3)				
$n_{1}$	$n_{2}$	$n_{3}$	Log-likelihood ratio	Proposed
5	5	5	0.1118	0.0480
5	8	15	0.0986	0.0564
8	12	16	0.0758	0.0486
10	15	20	0.0656	0.0460
30	30	30	0.0618	0.0550
Case (4)				
$n_{1}$	$n_{2}$	$n_{3}$	Log-likelihood ratio	Proposed
5	5	5	0.1088	0.0522
5	8	15	0.0962	0.0536
8	12	16	0.0714	0.0474
10	15	20	0.0680	0.0522
30	30	30	0.0562	0.0514

**Table 5 entropy-25-00111-t005:** Observed proportion of samples (in percentage) that was rejected homogeneity of means at 5% significance levels of 5000 simulated samples.

Case (5)						
$n_{1}$	$n_{2}$	$n_{3}$	$n_{4}$	$n_{5}$	Log-likelihood ratio	Proposed
5	5	5	5	5	0.1450	0.0530
5	7	8	10	15	0.0944	0.0502
8	12	16	22	30	0.0706	0.0478
10	13	16	21	25	0.0744	0.0508
30	30	30	30	30	0.0562	0.0488
Case (6)						
$n_{1}$	$n_{2}$	$n_{3}$	$n_{4}$	$n_{5}$	Log-likelihood ratio	Proposed
5	5	5	5	5	0.1320	0.0464
5	7	8	10	15	0.0980	0.0504
8	12	16	22	30	0.0826	0.0560
10	13	16	21	25	0.0768	0.0550
30	30	30	30	30	0.0584	0.0512
Case (7)						
$n_{1}$	$n_{2}$	$n_{3}$	$n_{4}$	$n_{5}$	Log-likelihood ratio	Proposed
5	5	5	5	5	0.1384	0.0462
5	7	8	10	15	0.1034	0.0480
8	12	16	22	30	0.0774	0.0514
10	13	16	21	25	0.0804	0.0544
30	30	30	30	30	0.0632	0.0544
Case (8)						
$n_{1}$	$n_{2}$	$n_{3}$	$n_{4}$	$n_{5}$	Log-likelihood ratio	Proposed
5	5	5	5	5	0.1362	0.0482
5	7	8	10	15	0.1014	0.0478
8	12	16	22	30	0.0750	0.0534
10	13	16	21	25	0.0754	0.0564
30	30	30	30	30	0.0576	0.0492

## Data Availability

Data is contained within the article.

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
