# Peer review of "Comparing Several Gamma Means: An Improved Log-Likelihood Ratio Test"

_entropy, 2023, doi:10.3390/e25010111_

Round 1

Reviewer 1 Report

In this paper, the author gives a Bartlett-type corrected log-likelihood ratio statis- tic, where the mean of the log-likelihood ratio statistic is estimated numerically by Monte-Carlo-type approach. The method can be applied to general test problem of simple or composite parametric hypotheses for specified parametric probability densities. The presented simulation results show that the proposed Bartlett-type corrected log-likelihood ratio statistic improves significantly the classical one in the sense that the approximation of the distribution of the proposed statistic by chi- square distribution is very accurate even when the sample size is small. The paper is well-written, and the result is of interest. Therefore, I recomand the publication of the paper. At the same time, I think that it will be of interest if the author investigates the power of the corrected ratio statistic, and to compare it with that of the classical one, at least by simulation.

Minor comment :
1. P. 1, L. 1 : Write “log-likelihood” instead of “log-ikelihood”.

Author Response

Thank you for the review.  The typo is fixed.

Reviewer 2 Report

The paper presents an applied statistical method for testing hypotheses about means for the gamma distribution based on the Bartlett correction method. The statement of the problem is given in the manuscript quite fully. The method is described in detail. The simulation results are also presented. The manuscript is rather short and does not present any new theorems. However, in general, all studies performed and described correctly. All the basic requirements for papers submitted to the journal "Entropy" are met.

There are a few minor comments:

1. I recommend use abbreviation "MLE" instead of "mle".

2. References should be extended. Please, cite some papers or books for the Wald, Wilks or log-likelihood ratio methods, etc. (see Introduction).

3. Are there any restrictions on the probability families for which this technique will be effective? What other types of hypotheses can be tested?

Author Response

Thank you for the review.

  1. I recommend use abbreviation “MLE” instead of “mle”.

Done.

  1. References should be extended. Please, cite some papers or books for the Wald, Wilks or log-likelihood ratio methods, etc. (see Introduction).

The references added are the books by Cox and Hinkley (1979), and Barndorff-Nielsen and Cox (1994).

  1. Are there any restrictions on the probability families for which this technique will be effective? What other types of hypotheses can be tested?

The Wald and log-likelihood ratio methods are standard asymptotic methods used in statistical inference.  As long as the likelihood function is available, with the regularity conditions stated in the two reference books, these methods can be applied.  In general, for papers that applied these methods, the regularity conditions are rarely stated. However, in a mathematical or theoretical papers, they will be stated.  The theme of this paper is more on applications of these methods, and this is why I did not explicitly state the conditions Nevertheless, in the revised paper, I have stated that "with the regularity conditions stated in Cox and Hinkley (1979), and also Barndorff-Nielsen and Cox (1994), ..." so that readers are aware that there are conditions for the methods to be applicable.